# Economic Valuation of the Renewal of Urban Streets: A Choice Experiment

**Yuhan Shao [1], Xinyu Xu [1,\*], Like Jiang [2] and Romain Crastes dit Sourd [3]**

[1]  Department of Landscape Architecture, College of Architecture and Urban Planning, Tongji University, Shanghai 200092, China; shaoyuhan@tongji.edu.cn
[2]  Institute for Transport Studies, University of Leeds, Leeds LS2 9JT, UK; L.Jiang2@leeds.ac.uk
[3]  Leeds University Business School, University of Leeds, Leeds LS2 9JT, UK; R.CrastesditSourd@leeds.ac.uk
[\*]  Correspondence: 1930125@tongji.edu.cn

**Abstract:** Examination of users' preferences and needs can provide an additional and strong basis for decision making, which is applicable in the case of urban street renewal. In this study, a choice experiment on street renewal plans in Shanghai was conducted using an online survey (N = 546), and people's willingness to pay (WTP) for a set of street attributes was estimated, including bicycle lane separation, pedestrian path width, green looking ratio and recreational and commercial amenities. By comparing WTPs, the results show that people had greater preferences for adding resting facilities than any other attributes in this scene, and they also give some examples of prices of street attributes in a street renewal scene. The gender, age and occupation of participants had a significant effect on WTPs. Females showed greater WTP for setting separate bicycle lanes and improving greening and amenities, and the age of street users had a positive effect on WTP through the payment for street renewal. The reasoning section of the survey indicated the concern on the cost–benefit ratio, the need to renew and overall impression when choosing, and only a few participants were unwilling to pay anything for street improvement. This kind of economic valuation can estimate the values that people place on street attributes that are otherwise not measurable in design and planning practice; it can help us understand public preferences for street renewal and support decision making.

**Keywords:** street renewal; stated preference method; choice model; willingness to pay; valuation

## 1. Introduction

After rapid expansion in the past two decades, cities in China such as Shanghai face the following problem: while the population is gathering in the city centre, it is also accompanied by the phenomenon of suburbanisation, and the need to seek the way of urban land stock development, which indicates the overwhelming trends of urban regeneration [1]. In this context, the formalistic and utilitarian approach of space making has been mainstream for a long time, such as the reconstruction of old areas orientated by land economy in Guangzhou [2] and Shenzhen [3]. However, this kind of renewal dominated by the reconstruction of buildings is far from the environment needed to a truly liveable city that brings a real estate boom, while gentrification has become one of the new problems that cannot be ignored. Thus, the renaissance of public spaces that primarily contain streets and squares aimed at a higher quality of urban life starts to attract more attention where street space plays a significant role.

Exploration and reflection on urban construction patterns have presented many perspectives since the first urban design conference was held at Harvard University in 1956 [4], one of which mainly emphasised the importance of walkability for truly high-quality urban spaces. Studies on walkable cities have never stopped [5,6]. Southworth [7] put forward six walkable urban design criteria that focused on the social attributes of the street space in addition to the physical quality of the space,

and the study also proposed the role of public awareness, noting social attributes and that a walkable street is not just physical space. That is, different groups of people have different understandings and preferences about the street environment. However, a limited change of street space requires consideration of the needs and preferences of street users. These attributes are aimed at pedestrian and bicycle traffic [8] and include street appearance and the supply of social activity places [9]. They also include the improvement of urban ecosystems [10,11].

As with subjective choice based on self-orientation by stakeholders, economic valuation using the stated preference (SP) method has become a feasible means and potential for environmental valuation [12]. It shows the preferences of users or stakeholders and provides a relative reference by introducing the concept of willingness to pay (WTP) for non-market goods, the value of which cannot be easily calculated. Most of the existing research covers many fields, such as transport and street greening. For example, there have been studies on the most important attributes of shared streets from the perspectives of pedestrians and drivers [13], and the preference for different degrees of street greening in which WTP was investigated [14–16]. There has also been a study that analysed preferences and attribute utilities of street greening without considering the WTP [17], which provided less direct evidence than those with economic valuation. In terms of the regeneration of the urban realm, several studies used the SP method and achieved significant results. A representative one is Atkins and Institute for Transport Studies (ITS) [18], which used a choice experiment for the evaluation of urban landscapes in four cities in the UK, focusing on pedestrian environments including attributes of street pedestrian mode, activity and surface material of the pedestrian path. The study was commissioned by the UK Department for Transport to explore potential approaches to value the benefits of pedestrianisation or townscape improvements. In general, it is difficult for a single study to cover all the attributes of a street due to contradiction between the limited number of attributes that can be measured simultaneously and the multiple and complex attributes of the street space. The more attributes considered, the larger sample size modelling will require; the model would hardly fit well. The research that considered large number of street attributes also tended to classify those attributes for several packages. For example, Sheldon et al. [19] investigated 15 attributes of street improvement in London and broke them down into three packages with each five attributes and then linked them with an extra model. Subsequent studies rarely involve so many street attributes. For example, Nellthorp et al. [20] focused on five street attributes and used a two-level method combining stated preference and a priority ranking approach. The role of the above studies is not just to calculate the WTP of attributes of a particular local street; these original studies provide the necessary metadata for the integration of benefit transfer (BT) methods into a larger scale, given the need for extensive, complex and standardised original studies to improve the accuracy of BT [21]. However, such research is still very limited in literature, and, to the best of our knowledge, no such research had been done in China. On the basis of existing studies, this study aims to further explore people's preferences for the improvement of street environment in the context of payment method in China. At the same time, different groups' preferences are also the focus of this study.

Using a choice experiment via an online survey, the objectives of this study are:

- Analysing people's preferences for different street attributes in urban street renewal and estimating their WTP for these attributes;
- Examining the effects of gender, age and occupation on people's preferences; and
- Exploring the feasibility of applying the choice model method to the public opinion survey for street renewal.

Section 2 of this paper describes, in detail, the experiment design and implementation. Section 3 presents the results in response to the study objectives, and Section 4 discusses the results. Section 5 concludes the paper.

## 2. Methods

In order to study people's preferences for street renewal for pedestrian and bicycle traffic, street greening, street leisure and commercial amenities, and the relationship between social demographic attributes and choice behaviour, a choice experiment was conducted and people's willingness to pay (WTP) for these street attributes was estimated (Figure 1).

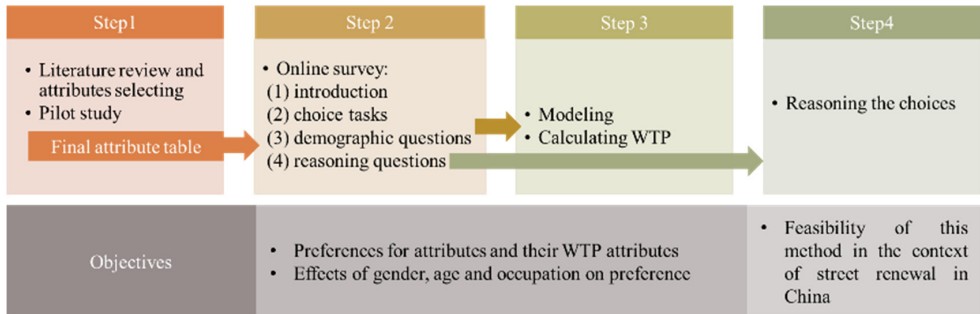

**Figure 1.** Study protocol.

### 2.1. Attributes of Street Renewal

Based on the four aspects of the investigation that came from previous references, a pilot study was conducted, which concerned excess attributes, such as the distance of the street crossing, pedestrian crossing facilities, style of vegetation, lighting and so on. However, some attributes showed collinearity and obstructed the efficiency of single choice behaviour in weighing attributes due to their incompatibilities with the scale of the scene set by the survey. Moreover, it took about ten minutes to complete the questionnaire in the pilot study due to the redundancy of attribute images, while it took about four minutes in the formal experiment. And the sample size required for modelling in pilot study was too large.

After several trials, five attributes were finally used in the formal experiment. They are all physical and specific and often appear in street renewal in China. Considering the influence of collinearity between attributes on model analysis, the independence of attributes was given special attention. For slow traffic, the available pedestrian path width and independent bicycle path were set, which were different from concrete attributes for which there were a number of choices, such as paving and guardrails, but pedestrian and bicycle paths only support limited choices and are relatively mandatory in street design. In terms of street greening, the attribute of the green looking ratio, which refers to the proportion of green elements in the field of vision, is determined due to the diversity of street greening configuration that affects choice behaviour more complexly [16]. In terms of amenities, the street furniture for rest and the shops facing the street represent functions of leisure and commerce, respectively. Finally, a change in rent due to the street updates was set as a proxy to calculate the WTP but did not represent actual implementation costs for the selected attributes. The five street attributes are categorical variables (and the dependent variables of model), with two or three levels. Level one of all attributes is the assumed street status quo that needs to be updated, whose combination forms the reference group in the choice experiment. Three levels of monthly payment options were used instead of exact amounts to reduce the complexity of the choice experiment for the participants. The settings for the attributes are shown in Table 1.

**Table 1.** Variable attributes and their levels of the choice experiments.

| Attribute | Code | Level 1 | Level 2 | Level 3 |
|---|---|---|---|---|
| Bicycle lane | BL | not separate | separate | - |
| Pedestrian path width (accessible) | PW | <1.5 m | 1.5–3 m | 3–8m |

**Table 1.** *Cont.*

| Attribute | Code | Level 1 | Level 2 | Level 3 |
|---|---|---|---|---|
| Green looking ratio | GL | <20% | 20%–35% | >35% |
| Street furniture | SF | no | only seats for bus waiting | with other furniture (for sitting, etc.) |
| Commercial amenities | CA | few | adequate | - |
| Monthly payment | P | 0 | low (100 RMB) | high (200 RMB) |

### 2.2. Choice Experiment

In the choice experiment, a scene of a street renewal was adopted in which the participants imagined themselves as the users of a street that was significant to them for commuting. Three alternative renewal scenarios were given, with the third being a reference group with the street left unchanged. The participants were asked to choose which one of three scenarios they preferred. According to this scene, the orthogonal design (see Table A1) of SPSS gave 27 cards when three scenarios were the same in Card 1, which was eliminated in the experiment.

In fact, there are far more than 27 combinations of scenarios, but the orthogonal design greatly reduced the number of experimental tasks and hence the sample size needed, which basically covers most of the combinations in reality. However, the results of this experiment cannot be used to reflect the choice preference of a particular combination of attributes but indicates the choice preference of each attribute.

The attributes of the choice experiment are illustrated with pictures and simple words to avoid the irritation and incomprehensibility brought about by a literal explanation of terminology (a choice task sample is shown in Figure A1). Image sources included Daxue Road, Guokang Road and Sujiatun Road in Shanghai, but they look very ordinary and contain no regional features of the type of street that can be found in most cities in China. Different attributes are not presented from the same perspective, but they all aim to highlight the differences between attributes and different levels. All the pictures used in the survey are shown in Appendix B.

### 2.3. Online Survey Design

The experimental subject is a hypothetical urban street renewal project in Shanghai. As one of the largest cities in China, Shanghai has a great demand for smart growth and the improvement of urban space quality, which indicates the inevitability and wide public acceptance of street renewal in this city. The survey was conducted on an internet questionnaire platform (Tencent Questionnaire) and collected in the form of an anonymous questionnaire. In order to further demonstrate and explore the influence of respondents' professional backgrounds on choice preference in the choice model and WTP analysis, the survey controlled the proportion of respondents with or without a professional background to make it approximately equal.

The electronic questionnaire used in the survey was divided into four parts. The first part described the situation of the street to be updated and the way the users (participants) use the street in detail, and set a specific situation for the choice task, so that the participants could have a greater sense of substitution. The choice experiments section asked the participants to make 13 decisions based on the random selection of 26 cards by the platform in each of which, pictures of the street view from a pedestrian or a cycling perspective were used to detail the differences between the various scenarios. The reasoning section used multiple choice questions to examine how the participants made their decisions and why some of them chose not to regenerate the street. The social demographic section recorded some basic information about participants, including gender, age and whether they were a practitioner of the built environment subjects. Screenshots of the online survey (Figure 2) show the interface for filling in the questionnaire on a mobile phone. Most of the questions were closed, and only the third section had a certain degree of openness to accept more free answers; but the proportion of respondents who answered freely was not high.

The survey was conducted after several rounds of testing and was improved in terms of comprehensibility and sensory comfort, in order to prevent respondents from receiving too much information to make a valid choice during completion. The final survey was conducted from 5th to 20th December 2019.

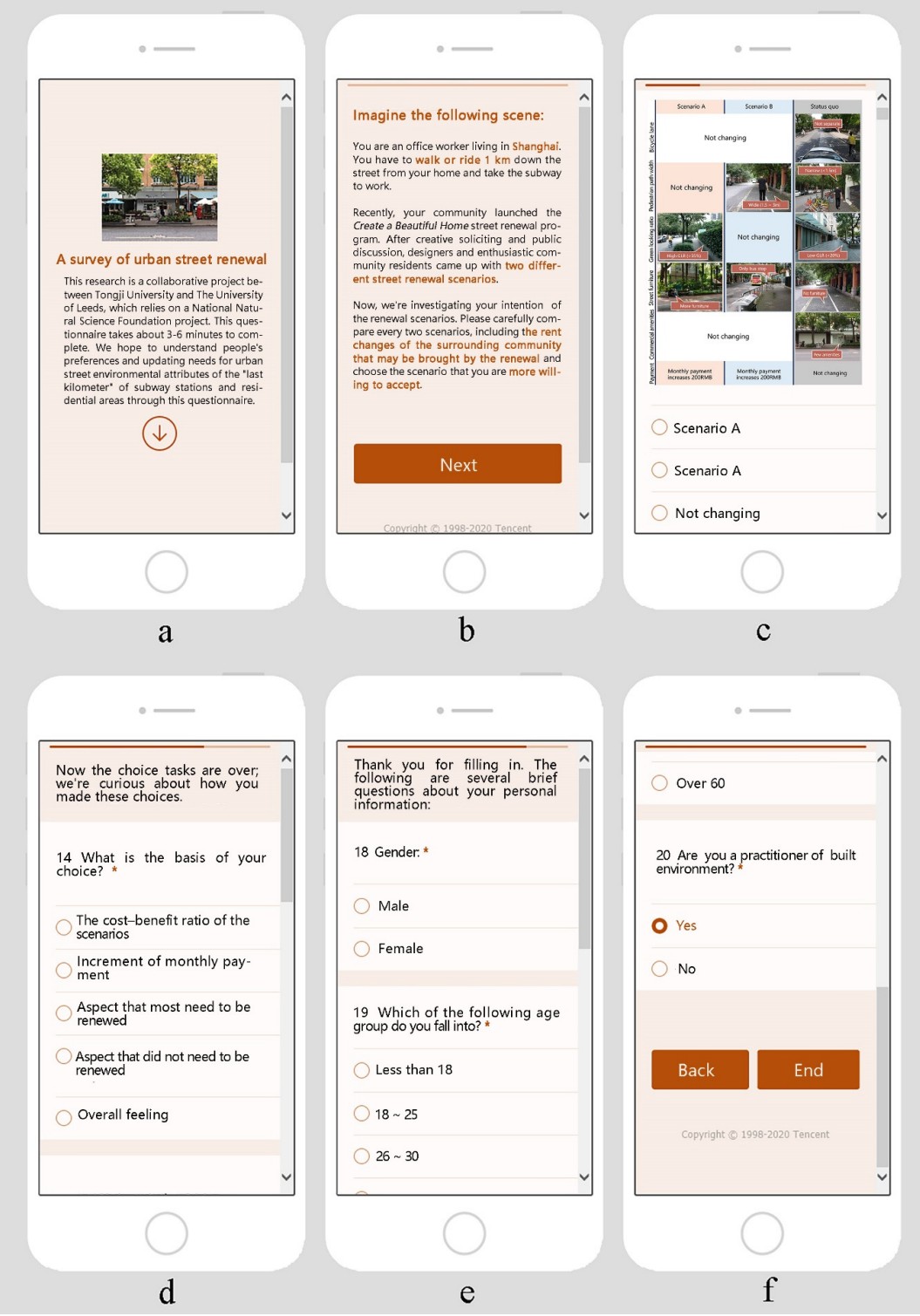

**Figure 2.** Screenshots of the online survey in English; however, only the Chinese version was used during data collection for this study: (**a**) introduction; (**b**) description of the hypothetical scene; (**c**) choice task; (**d**) reasoning questions; (**e**) demographic questions; (**f**) submission interface.

*2.4. Modeling and Calculation of WTP*

The data from the choice experiment were estimated by a multinomial logit model using Apollo, an R package for choice model estimation [22]. First, a model containing all the street attributes and price reference without considering the interaction between each attribute and each demographic variable was estimated (Model 1). Then trials were conducted to find out the significant (at $p < 0.10$) interaction variables to establish the second model (Model 2). Model 2 took the interaction between demographic variables and attribute variables fully into account and set age as an independent variable to represent the sensitivity of payment to age.

Although the model coefficients were statistically significant, the specific meaning and motivation of the choice behaviour cannot be explained by just examining the model parameters. As a monetised measure of values of non-market goods, WTP helps to further analyse the value or acceptability of each attribute. While the model parameters are clearly shown, an equation denotes the calculation method of the WTP of one level x (not the control level $x_{min}$) of the attribute t:

$$\text{WTP}_{t_x} = -\frac{\beta_{t_x} + \beta_{t_x} * interaction}{\beta_{payment} * \left(\frac{age}{mean\ age}\right)^{\beta_{age}}} \tag{1}$$

## 3. Results

*3.1. Response to the Online Survey*

A total of 546 valid copies of the questionnaire with 7089 responses to the choice experiments from a total of 642 respondents were collected in 15 days. Invalid copies from participants who spent less than 60 s and more than 1200 s to complete them were eliminated as a result of box plot analysis and a 209.5 s ± 453.0 s (mean ± standard error) average time.

Due to the universality of the research objects, the questionnaire did not set more limits on respondents; therefore, the respondents were from all the Chinese provinces and their ages were mostly between 18 and 50, which is the age group that can best adapt to using electronic questionnaires. Respondents aged 18 to 50 accounted for 95.3% (n = 642), which indicates that the results of this research can better reflect the choices of the young and middle-aged, while evidence is insufficient for children and the elderly. The sex ratio and the proportion of practitioners and non- practitioners are approximately equal. Table 2 details the information of the respondents of valid copies.

**Table 2.** Basic information table (n = 546).

| Category | | Frequency | Percentage |
| --- | --- | --- | --- |
| Gender | female | 322 | 58.97% |
| | male | 224 | 41.03% |
| Age | <18 | 8 | 1.47% |
| | 18–25 | 157 | 28.75% |
| | 26–30 | 169 | 30.95% |
| | 31–40 | 151 | 27.66% |
| | 41–50 | 39 | 7.14% |
| | 51–60 | 13 | 2.38% |
| | >60 | 9 | 1.65% |
| Practitioner | no | 301 | 55.13% |
| | yes | 245 | 44.87% |

*3.2. Model Parameters and Estimation*

All parameters were statistically significant at the 95% confidence interval in Models 1 and 2 except the interactions between gender and GL (l2) and SF (l2), and each attribute also displayed

heterogeneity, as indicated by statistically significant standard deviations. Table 3 shows the parameter estimates of two models for the valid response of choice experiment.

Model 1 shows that all the five attributes studied had a significant positive effect on the probability of a street renewal scenario being chosen, while the monthly rent changes had a significant negative effect, the extent of which will be explained more clearly in the following WTP analysis. Among the attributes of street renewal, it was clear that the addition of separate bicycle lanes and furniture for recreational streets was favoured by the participants. However, the effect on the increment of the green looking ratio was not so large, either from low to medium or from medium to high.

The trend of the main variable parameters in Model 2 is the same as that in Model 1. In previous trials, gender showed a significant influence on attributes, which suggested that females appeared to have a higher probability of choosing an updated attribute than the control level that would not change the street status quo. In addition, whether there was a professional background also had an impact on the preference for adding commercial facilities: professionals seemed to have more acceptance of adding stores. According to the research, price changes did correlate with age. This can be well explained by the income sensitivity of price and the collinearity of age and income.

The resulting model had an $R^2$-adj less than 0.2 (Mode 1 $R^2$-adj = 0.1321, Mode 2 $R^2$-adj = 0.1344), and values of $R^2$ between 0.2 and 0.4 are considered to be a good fit [23]. This shows that, although the coefficients of all parameters were significant in Model 1, the whole model still did not fit well. When it is used for prediction, the error may be quite large. However, the model still makes sense because the research focuses on individual attributes rather than prediction with the integration of street scenario.

**Table 3.** Parameter estimates for a multinomial logit model.

| Variable | Model 1 | | | | Model 2 | | | |
|---|---|---|---|---|---|---|---|---|
| | Coeff. | | Std. Err. | t-Ratio(0) | Coeff. | | Std. Err. | t-Ratio(0) |
| main variable | | | | | | | | |
| BL | 0.414 | ** | 0.036 | 11.586 | 0.325 | ** | 0.054 | 6.03 |
| PW (l2) | 0.308 | ** | 0.042 | 7.279 | 0.311 | ** | 0.042 | 7.32 |
| PW (l3) | 0.233 | ** | 0.042 | 5.541 | 0.237 | ** | 0.042 | 5.61 |
| GL (l2) | 0.222 | ** | 0.042 | 5.234 | 0.190 | ** | 0.062 | 3.07 |
| GL (l3) | 0.278 | ** | 0.042 | 6.640 | 0.160 | ** | 0.062 | 2.60 |
| SF (l2) | 0.337 | ** | 0.043 | 7.896 | 0.262 | ** | 0.063 | 4.18 |
| SF (l3) | 0.494 | ** | 0.043 | 11.597 | 0.391 | ** | 0.062 | 6.29 |
| CA | 0.276 | ** | 0.037 | 7.567 | 0.199 | ** | 0.0479 | 4.15 |
| P | −0.00224 | ** | 0.000363 | −6.175 | −0.00237 | ** | 0.000343 | −6.89 |
| interaction | | | | | | | | |
| female*BL | | | | | 0.157 | * | 0.070 | 2.254 |
| female*GL (l2) | | | | | 0.058 | | 0.077 | 0.750 |
| female*GL (l3) | | | | | 0.201 | ** | 0.077 | 2.611 |
| female*SF (l2) | | | | | 0.130 | | 0.078 | 1.665 |
| female*SF (l3) | | | | | 0.175 | * | 0.077 | 2.282 |
| practitioner*CA | | | | | 0.172 | * | 0.069 | 2.487 |
| $\beta_{age}$ | | | | | -0.731 | ** | 0.155 | -4.703 |

\* $p < 0.05$, \*\* $p < 0.01$.

## 3.3. WTP

The WTP calculated according to Formula (1) is shown in Table 4. The trend of WTP is consistent with the model coefficients. Through monetisation, it can be seen that the highest WTP to increase street leisure furniture was more than three times that for the slight increase in greening. The latter was approximately equivalent to the lowest acceptable value of monthly rent set in the experiment. The former, however, far exceeded the monthly rent variation, suggesting that the addition of street furniture was somewhat acceptable, even at higher prices, which reflects the scarcity of leisure furniture in the current street environment and people's need for it. In the case of attributes with three levels, the WTP decreased when the available pedestrian path width was increased to a higher level,

which indicates that respondents may have had a particular preference for pedestrian path width, which is most likely between 1.5 and 3 m.

The WTP of this study showed a relatively obvious gender correlation, with the WTP of females being generally higher than that of males, especially in the aspect of increasing greening. For females, higher levels of the green looking ratio were significantly more acceptable, while, for males, it was less acceptable. And males had the lowest WTP increase of all attributes.

In this study, WTP was significantly affected by age: from 18 to 50 years old, the higher the age, the greater the WTP of each attribute. The influence of age on WTP is common in this study and in other studies on urban landscapes, including urban forest tourism [24] and green infrastructure [25].

**Table 4.** Monthly willingness to pay (WTP) of each attribute (unit: RMB).

| Variable | Common WTP | Category | WTP of Given Age | |
| --- | --- | --- | --- | --- |
| | | | 25 | 50 |
| Bicycle lane | 184.69 | male | 117.80 | 195.52 |
| | | female | 174.72 | 290.00 |
| Pedestrian width (1–2) | 137.57 | | 112.75 | 187.15 |
| Pedestrian width (2–3) | 104.13 | | 85.89 | 142.56 |
| Pedestrian width (1–3) | 241.70 | | 198.64 | 329.71 |
| Green looking ratio (1–2) | 99.12 | male | 68.92 | 114.22 |
| | | female | 90.01 | 149.16 |
| Green looking ratio (2–3) | 124.24 | male | 57.93 | 96.16 |
| | | female | 130.73 | 216.98 |
| Green looking ratio (1–3) | 223.36 | | | |
| Street furniture (1–2) | 150.62 | male | 94.8 | 157.36 |
| | | female | 141.92 | 235.56 |
| Street furniture (2–3) | 220.34 | male | 141.89 | 235.51 |
| | | female | 205.23 | 340.64 |
| Street furniture (1–3) | 370.97 | | | |
| Commercial amenities | 123.38 | practitioner | 134.31 | 222.92 |
| | | non-practitioner | 72.13 | 119.72 |

### 3.4. Analysis of Reasoning

In addition to the interpretation of the model coefficients and WTPs, the analysis of the data collected from the reasoning section of the questionnaire also explains the choice experiments, especially in terms of the motivation to choose or not to choose. A one-choice question asked the participants to select the main basis for 13 choice experiments (the results are shown in Table 5), followed with a fill-in question to supplement other possible bases. The given options summarised most reasons and it was seldom that participants filled in the blank. Concerning the setting of the control scenario, the reasons why participants did not choose to change may provide an explanation of the feasibility of the attribute setting. An average 16.23% probability of the control scenario justified the indispensable reasoning to further analysis.

**Table 5.** Main basis for 13 choice experiments (n = 543, 3 participants did not answer this question).

| Category | Percentage |
| --- | --- |
| The cost–benefit ratio of the scenarios | 46.78% |
| Overall feeling | 21.73% |
| Aspect that most need to be renewed | 18.78% |
| Increment of monthly payment | 10.31% |
| Aspect that did not need to be renewed | 2.39% |

Only 25 participants (6.19%; see Table 6) showed an unwillingness to pay for the street renewal, while others paid more attention to the cost–benefit ratio of regeneration and even gave some useful advice on the attributes through the fill-in question. That is to say, most people had a desire for the improvement of street space, including easier access to walking and cycling, more contact with nature and for more amenities, despite a slight but acceptable cost to it. Additionally, a 46.78% focus on the cost–benefit ratio suggests that the cost–benefit ratio may have a positive effect on choice behaviour.

**Table 6.** Reasons for not changing (n = 404, multiple-choice question).

| Category | Percentage |
| --- | --- |
| Payment is too high | 51.24% |
| The cost–benefit ratio of A and B are poor | 42.33% |
| Changing makes no sense for me | 32.43% |
| Aspect that does not need to be renewed | 17.82% |
| Decision is too difficult to make | 10.15% |
| I do not want to pay any money for street renewal | 6.19% |

The cost–benefit ratio is very similar in nature to the WTP, which is the ratio of utility to price and more for non-market goods. In existing research, WTP can act as a bridge to connect and compare hard-to-relate attributes, such as biodiversity and employment [26]. In this study, WTP also plays such a role due to the attributes used belonging to the four aspects of street renewal. Direct comparison and balance between attributes seem to be difficult, and the choice of combination is not as directly acceptable as the result of each attribute itself.

It was also important to note that more than half of the participants who chose not to change the status of the street in the choice task cited higher rent charges as the main reason. This was confirmed in another previous choice experiment study of watershed restoration [27]. Similar to its experimental results, despite the high price level reported in this study, no participant had consistently chosen the unchanged scenario. On the one hand, it shows the validity of the data. On the other hand, it also makes the WTP likely to be much greater than people would be willing to pay in practice, which fits nicely with the conclusions of a monetisation study of greenways [28], suggesting that the WTP may represent some kind of vision, rather than a precise representation of real value.

## 4. Discussion

### 4.1. Preferences on Four Aspects of Street Renewal

The choice experiment conducted in this study reflected the different concerns and preferences of participants from all over China over slow traffic environments, greening and leisure and the commercial amenities of streets in a street renewal project and showed that the preferences were influenced by non-street attributes, including the user's age, gender and professional background, which had not been taken into account in previous studies.

From the perspective of modes of transport, a slow traffic environment includes walking and cycling space. In this study, participants had a certain preference for allocating the space of these modes of transport, compared with streets that are not suitable for them. However, the attributes of slow traffic in this study were relatively basic, considering the feasibility of slow traffic but not its comfort, which, in itself, significantly affects the path choice [17].

The results of this choice modelling strongly support the previous research on street greening and show the significance of people's preferences for increasing street greening. The study also separately reported a clear preference for street greening among females, compared with males. The WTP for greenness is not the same as that of the study of Giergiczny et al., which showed that the scarcer street greening is, the higher the WTP for increasing greening will be [15]. In this study, this was true for

the male participants, but the opposite held for the female participants. This may also be explained by there being no 'no greening at all' in the attribute level, and the magnitude of the rate change is indeed relatively ambiguous. Nevertheless, the study supports people's preferences for advanced landscaping configurations [16] because high greenness cannot be derived from simple configurations such as ground cover and hedges, where trees are required.

The results of the study reported that street leisure amenities had the highest WTP among all attributes. This seems to contradict the scene of choice experiment viewing the street as the only way to commute in the experimental setting; or it can be understood as people's preferences and yearning for free leisure time and comfort itself. This indirectly shows that street renewal is not limited to traffic space but is relevant to the pursuit of enhanced quality of life and corresponding lifestyle.

One particular result of this study showed a clear preference for increasing street business facilities in participants with a professional background, with WTP nearly twice as high as that of non-professionals. For commercial amenities, the choice of location depended to some extent on the balance between the store owner and public policy, which indicates that planning needs to be part of the process [28]. The low receptivity of non-professionals to commercial amenities may be due to their insufficient understanding of public policy mechanisms. Setting the attribute of commercial amenities is a new trial of this study, which intends to expand the attention to the material attributes of the street itself to explore the atmosphere and amenities of the street that reflect the street value of urban realm as a whole [29], and may further expand to a wider range of communities and blocks. The attention to the participants' professional background was the first of its kind in the study; and it showed the attribute with discrepancy between public opinion and designer intent, suggesting careful consideration in determining design scenarios.

In general, the choice model and WTP method used in the study have basically achieved the objectives and clearly reflect people's preferences and economic acceptance of various attributes of street renewal. Consideration of social attributes further refined these preferences so that the results reflected the differences among different groups; the focus on different groups of people is one of the advantages of this study. The re-examination of the choice behaviour in the reasoning section helps explain the motivation and enables researchers to analyse the meaning behind the numbers. This approach is feasible to a certain extent, and may have important potential in opinion polls for public processes such as street renewal, which is based on now widely used virtual technologies and electronic platforms, greatly reducing the manpower cost of SP methods. The appropriate use of electronic methods and the setting of virtual scenes is another advantage of this study, which can break through the regional limits to a certain degree and is conducive to the study of more general problems. This approach is also quite adaptable for the study of other specific types of space, such as urban green space [30]. However, it may be necessary to choose an appropriate approach for the locality of the research problem itself because, as has been shown in many studies using the contingent valuation method, the WTP is influenced by participants' cognition and location [31,32], and directional heterogeneity is significant in distance decay [33].

*4.2. Use of the Results for Decision Making*

The ultimate goal of this study is to use the modelling results and the value of WTP to help decision making. Output of Atkins at el. [18] in the UK was adopted by the Department for Transport as a useful tool and can be used on local cases, although not mandatory [34]. Previously, a report by Chartered Association of Building Engineers (CABE) [35] on the value that good design created for streets showed the retail and house price growth that good street design brought to ten high streets in London; the report divided these values into user benefits and market prices, which were calculated using the stated preference method and revealed preference method, respectively. In China, economic benefit is an important aspect of the evaluation of a single business case. Engineering cost method is commonly used, while the SP method, especially the choice experiment method, has no precedent before. To some extent, this may lead to the underestimation of the user benefits.

When it comes to public events such as street renewal, it is crucial to consider user benefits. Now the common way to improve user benefits is to let stakeholders participate in the forum discussion and community interviews in the early stage of a project. This collection of public opinion is obviously inadequate. With the SP method, especially the choice experiment method, the stakeholders firstly have the right to choose, and can learn more about the different combination forms of attributes and possible improvement through a series of choice tasks. Although individual choice does not equal the final solution, the group's preference clearly provides the designer with the necessary reference of design specification. When integrating the preference information of different groups, weighting according to the proportion of different groups in the target region is a method that can be considered; and it can be assigned different weights to different groups when modelling. These considerations can be applied to the actual situation on a case by its basis.

In addition to using modelling and WTP results to assess economic benefits and demonstrate the feasibility of design, they have the potential to develop street design guidelines [18] and identify streets that need to be renewed that may be considered in further research.

### 4.3. Limitations

However, there are some limitations to this study. Of course, the slightly low fit of the model is one of them, which indicates the unreliability of the model when used for prediction. The result is qualitative rather than quantitative, which is related to the fact that the attributes used are categorical attributes and indicates that there may be some contradiction between the efficiency of participant choice behaviour and the accuracy of variables.

Another limitation is the value of the WTP. A previous study by Sheldon et al. [19] explored the impact of the use of council tax, rent and public transport fares as cost measures to calculate the WTP; there is no council tax in China; and public transport fares do not change in response to market conditions. Therefore, this study used monthly rent increment as a measure of WTP. As the scene of choice experiment was set in Shanghai, one of the cities with the highest housing price in China, when the reference variable is the monthly rent, it is undoubtedly affected by the comparison effect, which makes it seem more influential. This makes the WTP reported in this experiment very high. This may also be related to the conclusion of research on park ticket prices in Guangzhou, which inferred that Chinese people have little understanding of various payment methods and lack an economic concept of public goods [36]. Combined with a review of research on the monetisation of urban green space, it can be found that although the WTP has many advantages as a scale for the valuation of unquantifiable public goods, it should not be attributed to the economic benefits of the motivation of the improvement of public space [37]. WTP is not equal to the actual currency value [14]. However, more metadata are required to further the monetisation of street renewal in the context of Chinese payment methods.

## 5. Conclusions

In this study, a street renewal choice experiment using an online survey with simulated scenarios reported the preferences of 546 participants. The analysed results of the choice model and WTP show that:

- There are significant and different preferences for the slow traffic environment, greening and leisure and business amenities in street renewal; people have the greatest preference for adding resting facilities and the least for adding commercial facilities;
- Demographic attributes of participants, such as gender, age and whether they have professional backgrounds, significantly influenced their preferences for street renewal. Females showed greater WTP on separate bicycle lanes, street greening and leisure amenities than males; and the age of participants had a positive effect on WTP, while the participants with professional backgrounds also showed a preference for adding commercial amenities, which is a particular result in this study;

- The results of WTP are similar to the cost–benefit ratio that people pay attention to, but at the same time, they are not all accurate but served as the symbols of public vision in the renewal project, given that the alternatives of proxy variables may be affected by urban consumption. The SP method and WTP are still feasible for studying public events and products and giving evidence of public preferences, such as in street renewal in the context of China.

Although the cost of street renewal can be precisely calculated in a practical project, the improvement to the attributes of street renewal cannot be described clearly in respect of their benefits for street users. However, an examination of WTP reflects a type of economic valuation that is entirely based on the preferences and needs of users expressed by their choice behaviour in a hypothetical scene. The economic valuation can provide feasibility for decision makers to make comparison with the project cost. And it may be noted that the value that people place on each street attribute is hardly relevant to the project cost but directly related to their preferences and needs, which makes great sense of decision-making to some extent. The direction of our future research is to apply the choice experiment method to the identification of the streets and their attributes worth updating and the valuation of scenarios in urban street renewal in China.

**Author Contributions:** Conceptualization, Y.S. and L.J.; methodology, L.J. and R.C.d.S.; software, R.C.d.S.; validation, Y.S., X.X., L.J. and R.C.d.S.; formal analysis, X.X., L.J. and R.C.d.S.; investigation, Y.S. and X.X.; resources, Y.S. and X.X.; data curation, X.X.; writing—original draft preparation, X.X.; writing—review and editing, Y.S., X.X. and L.J.; visualization, X.X.; supervision, Y.S.; project administration, Y.S.; funding acquisition, Y.S. All authors have read and agreed to the published version of the manuscript.

**Funding:** This research was supported by National Natural Science Foundation of China (NSFC) "Urban Natural Landscape Visual Comfort Mechanism Research" (No. 51808393).

**Acknowledgments:** The authors would like to thank all the participants who took part in the online survey.

**Conflicts of Interest:** The authors declare no conflict of interest.

## Appendix A. Orthogonal Design and a Sample of Cards

**Table A1.** Orthogonal design of choice experiment.

| Card Number | BL-A | PW-A | NL-A | SF-A | CA-A | P-A | BL-B | PW-B | NL-B | SF-B | CA-B | P-B |
|---|---|---|---|---|---|---|---|---|---|---|---|---|
| 1 * | 1 | 1 | 1 | 1 | 1 | 100 | 1 | 1 | 1 | 1 | 1 | 100 |
| 2 | 2 | 1 | 2 | 2 | 1 | 100 | 2 | 1 | 2 | 1 | 1 | 100 |
| 3 | 1 | 3 | 1 | 2 | 1 | 100 | 1 | 1 | 1 | 3 | 2 | 100 |
| 4 | 2 | 2 | 1 | 3 | 2 | 200 | 2 | 1 | 1 | 2 | 1 | 100 |
| 5 | 2 | 2 | 2 | 1 | 1 | 200 | 1 | 3 | 1 | 1 | 1 | 200 |
| 6 | 1 | 2 | 1 | 3 | 1 | 100 | 2 | 3 | 3 | 1 | 2 | 100 |
| 7 | 1 | 3 | 1 | 2 | 2 | 200 | 1 | 2 | 2 | 1 | 1 | 200 |
| 8 | 1 | 1 | 2 | 2 | 1 | 200 | 2 | 2 | 3 | 2 | 1 | 100 |
| 9 | 1 | 1 | 2 | 2 | 2 | 100 | 2 | 3 | 1 | 3 | 2 | 200 |
| 10 | 1 | 2 | 3 | 2 | 2 | 100 | 1 | 1 | 3 | 2 | 1 | 200 |
| 11 | 1 | 3 | 2 | 3 | 1 | 200 | 1 | 1 | 2 | 3 | 1 | 100 |
| 12 | 1 | 2 | 1 | 3 | 1 | 100 | 2 | 2 | 2 | 3 | 1 | 200 |
| 13 | 1 | 3 | 3 | 1 | 2 | 100 | 2 | 2 | 1 | 1 | 1 | 100 |
| 14 | 2 | 3 | 1 | 2 | 1 | 100 | 1 | 3 | 3 | 2 | 1 | 100 |
| 15 | 1 | 1 | 1 | 1 | 2 | 200 | 1 | 3 | 3 | 3 | 1 | 100 |
| 16 | 2 | 2 | 3 | 2 | 1 | 200 | 1 | 2 | 1 | 3 | 2 | 100 |
| 17 | 1 | 1 | 3 | 3 | 1 | 100 | 1 | 2 | 1 | 2 | 1 | 100 |
| 18 | 1 | 2 | 2 | 1 | 1 | 100 | 1 | 2 | 3 | 3 | 1 | 100 |
| 19 | 1 | 2 | 3 | 2 | 1 | 100 | 1 | 3 | 2 | 1 | 1 | 100 |

**Table A1.** *Cont.*

| Card Number | BL-A | PW-A | NL-A | SF-A | CA-A | P-A | BL-B | PW-B | NL-B | SF-B | CA-B | P-B |
|:-----------:|:----:|:----:|:----:|:----:|:----:|:---:|:----:|:----:|:----:|:----:|:----:|:---:|
| 20 | 1 | 3 | 3 | 1 | 1 | 200 | 2 | 3 | 2 | 2 | 2 | 100 |
| 21 | 2 | 1 | 3 | 3 | 2 | 100 | 1 | 3 | 2 | 3 | 1 | 100 |
| 22 | 1 | 1 | 3 | 3 | 1 | 200 | 1 | 1 | 3 | 1 | 2 | 200 |
| 23 | 2 | 3 | 2 | 3 | 2 | 100 | 1 | 2 | 3 | 1 | 2 | 100 |
| 24 | 1 | 2 | 2 | 1 | 2 | 100 | 1 | 1 | 2 | 2 | 2 | 100 |
| 25 | 2 | 1 | 1 | 1 | 1 | 100 | 1 | 2 | 2 | 2 | 2 | 200 |
| 26 | 2 | 3 | 3 | 1 | 1 | 100 | 2 | 1 | 3 | 3 | 1 | 200 |
| 27 | 1 | 3 | 2 | 3 | 1 | 100 | 1 | 3 | 1 | 2 | 1 | 200 |

* in this set scenario A and B have the same attribute level with reference group; thus, eliminated in choice tasks.

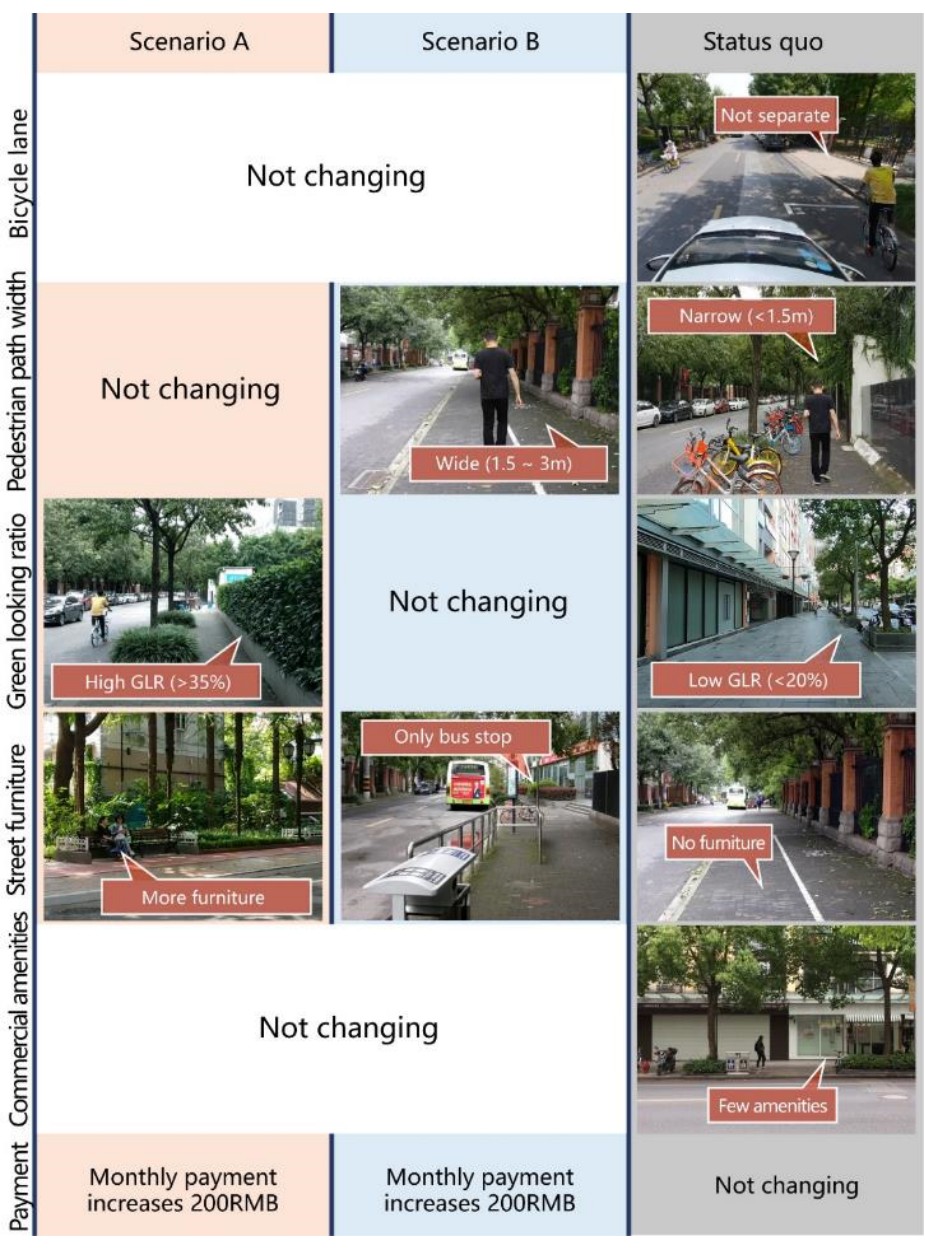

**Figure A1.** Sample of choice tasks (Card 17).

## Appendix B. The Attributes and Pictures illustrating Their Levels

**Table A2.** Attributes of street renewal and pictures illustrating their levels.

| Attribute | Level 1 | Level 2 | Level 3 |
|---|---|---|---|
| Bicycle lane | <br>not separate | <br>separate | - |
| Pedestrian path width (accessible) | <br>< 1.5 m | <br>1.5–3 m | <br>3–8 m |
| Green looking ratio | <br>< 20% | <br>20%–35% | <br>>35% |
| Street furniture | <br>no | <br>only seats for bus waiting | <br>with other furniture (for sitting, etc.) |
| Commercial amenities | <br>few | <br>adequate | - |
| Monthly payment | 0 | low (100 RMB) | high (200 RMB) |

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
