# Peer review of "Economic Valuation of the Renewal of Urban Streets: A Choice Experiment"

_sustainability, doi:10.3390/su12124808_

Round 1

Reviewer 1 Report

The paper is interesting and original, as it provides insights on the Economic Valuation of the Renewal of Urban Streets.

Below I would give some comments for better clarify the research path and structure:

1 Introduction

The theoretical background is suitable for the aim of the paper, but I would suggest to implement some key references on street renewal and stated preference method.

2 Methods

Materials and methods are clear and well explained, however, I would suggest inserting a research methodology explanation specifying each research step able in responding to the research objectives (a graphic on research methodology approach could help).

Reviewer 2 Report

The article concerns a interesting problem. It can be published after taking into account the following minor remarks:
1) line 80: please write something more about pilot studies.
2) Figure 1: Maybe for the purpose of this article you could create an English version of the application? In my opinion, the article should contain screenshots of the English version of the survey.
3) I think that what is in Figure 1 is not enough. I suggest to add it in section 2.2 or in the annex:
- pictures showing the streets that were presented to the respondents,
- pictures illustrating attributes in the application.

Reviewer 3 Report

Given that the purpose of the research is to conduct online survey, analyse its results and gain insight into the people’s willingness to pay (WTP) for a set of street attributes, it would be interesting to present papersvin the introduction section that also dealt with the collection of the same or similar information.

The paper deals solidly with the topic and its true classification would be the basis for future scientific research. For this reason, one part of it (I suggest a conclusion) should explain in more detail how the information gathered can facilitate decision-making in the future. How would the information gathered affect the final decision? What are the other stakeholders to consider when making a final decision? How could they act collectively? What methods would be used?

As already mentioned, the aforementioned similar researches is missing in the introduction. Given this, the discussion should provide a comparison of this research with the already existing one and emphasise how this research differs from the already existing ones. It is necessary to clearly emphasise the scientific contribution of the paper. It is also necessary to provide guidelines for future research. The paper deals solidly with the topic, but in order to maintain the scientific character, it is necessary to clearly state what step forward it offers in relation to existing research.

Round 2

Reviewer 3 Report

The authors have answered and included all the reviewer's comments in the manuscript. The manuscript is significantly improved. For this reason, I recommend to accept it as it is for the next step.